# Porphyry Copper: Revisiting Mineral Resource Assessment Predictions for the Andes

**Jane Marie Hammarstrom**

U.S. Geological Survey, Reston, VA 20192, USA; jhammars@usgs.gov

**Abstract:** A mineral resource assessment of porphyry copper deposits in the Andes Mountains of South America was done in 2005 in cooperation with geological surveys in South America. The study identified 590 million metric tons (Mt) of copper in identified resources. Continued exploration and development in the region over a 15-year period provide an opportunity to compare the predicted assessment results with new discoveries and resource growth in previously known deposits. The 2005 assessment estimated that 145 undiscovered deposits could contain a mean of 750 Mt of copper. The actual number of deposits increased (2005 to 2020) from 69 to 120 and the amount of identified copper resources increased from 590 Mt to 1600 Mt. Although most of the new deposits and copper resources are concentrated in Miocene-Pliocene and Eocene-Oligocene mineral belts, new deposits have been discovered in Jurassic and Cretaceous mineral belts. Resource growth in porphyry copper deposits known in the Andes in 2005 (1100 Mt copper) exceeds copper resources in new discoveries since 2005 (490 Mt copper) by a factor of 2.

**Keywords:** porphyry copper; Andes; mineral resource assessment

## 1. Introduction

Mineral resource assessments represent a synthesis of the state of knowledge about identified and potential undiscovered resources for a given deposit type in a given place at a given time. Ideally, mineral resource assessments are done on a recurring basis to incorporate new geologic data, improved digital map scales, and exploration results, evolving mineral system and mineral deposit concepts, and new assessment tools. The first global mineral resource assessment of porphyry copper deposits was completed over a period of more than 10 years using the U.S. Geological Survey (USGS) form of quantitative mineral resource assessment [1].

In this geology-based form of assessment, conducted at a scale of 1:1,000,000, geographic areas are delineated as permissive for the occurrence of porphyry copper deposits using criteria in descriptive mineral deposit models. Grade and tonnage models of deposits that have well-defined identified resources are used as analogs for the endowment of undiscovered deposits. Probabilistic estimates of numbers of undiscovered deposits assumed to share descriptive and grade-tonnage characteristics with those in the models are combined with grade and tonnage models in a Monte Carlo simulation to provide a mean and a distribution of estimated amounts of undiscovered copper resources at different quantiles. The global assessment was done for different world regions by the USGS in cooperation with many international institutions and individual collaborators. Regions were assessed in a series of studies published between 2008 and 2019. Results of the assessment of both porphyry copper and sediment-hosted stratabound deposits were summarized in a fact sheet [2], a GIS [3], and in a summary report and atlas [4].

Porphyry copper deposits represent the largest source of global copper supply, as well as significant sources of molybdenum, gold, and silver. Porphyry copper deposits occur in subduction-related magmatic arcs, in volcanic island arcs, and in belts of magmatic rocks that formed in postconvergent tectonic settings. They form at shallow crustal depths,

typically less than 4 kilometers (km), associated with calc-alkalic igneous rocks. Most known deposits are Mesozoic or Cenozoic in age although a few deposits as old as Archean have been reported [5–7]. Precambrian deposits are rarely preserved, which probably reflects the shallow emplacement of porphyry systems in tectonically active settings that are prone to uplift and erosion [5].

A porphyry copper assessment of the Andes region of South America was the first region assessed in 2005 and published in 2008 as part of the USGS Global Mineral Resource Assessment project [8,9]. The Andes host the largest known porphyry copper deposits in the world, and the region continues to be an area of active mineral exploration and development. Therefore, recent resource data for porphyry copper discoveries in the Andes provide an opportunity to compare what was predicted as of 2005 using this methodology with what has been discovered in the last 15 years. The 2005 Andes assessment was included in a global assessment of undiscovered copper resources, with the addition of an estimate of the amount of in-place undiscovered resources that might be economically recoverable [4,10]. Those data are also available in on-line [8] viewer at https://mrdata. usgs.gov/sir20105090z/map-us.html (accessed on 1 May 2022).

Although this form of quantitative mineral resource assessment was developed in the 1990s [11,12] and used in many USGS mineral resource assessments, this is the first retrospective study comparing probabilistic estimates of in situ undiscovered resources with new discoveries and resource growth.

## 2. Background: Porphyry Copper Assessment of the Andes

The 2005 porphyry copper assessment of the Andes identified 26 tracts of land that were deemed to be permissive for the occurrence Phanerozoic porphyry copper deposits [8]. Tracts were delineated on the basis of permissive geology at a scale of 1:1,000,000 assuming a depth cutoff to the top of a porphyry system of 1 km below the Earth's surface. Initially, tracts were constructed by considering island- and continental volcanic-arc subduction-boundary zones as the geologic environment permissive for porphyry copper formation and plotting locations of all known Andean porphyry copper deposits and prospects that were included in a global porphyry copper compilation [13]. Geologic map distributions of intermediate composition volcanic and (or) plutonic rocks that formed during periods of approximately coeval arc magmatism, as well as structural and tectonic controls, available geophysics and geochemistry, and regional expertise were used to refine tract boundaries and extend boundaries under shallow cover. Age ranges of host rocks and dated deposits were used to assign tracts to a geologic age (Figure 1). Tracts excluded areas that were considered too deeply eroded to host porphyry copper deposits. In some cases, coeval tracts were separated based on differences in level of erosion or extent of exploration. Permissive tracts ranged in size from about 2000 to 220,000 km$^2$ [8]. The original permissive tract designations used in the 2005 assessment (SA01PC–SA20PC) were renumbered in the global assessment using the permissive tract code adopted for that study (005 for UN region for South America, pCu for porphyry copper, and a 4-digit number starting with 1001). For example, tract SA01PC is the same as tract 005pCu1001, and so on (Table 1).

There were 69 discovered porphyry copper deposits identified in the study area at the time of the assessment in 2005 (Figure 1). The term "discovered", or "known" deposit was restricted to deposits reported in the literature to be well-explored in three dimensions and have publicly available ore tonnages and grades. Identified resources include past production, reserves, and measured, indicated, and inferred resources at the lowest cutoff grade reported. Other examples of porphyry-style mineralization that were not well-characterized were considered as prospects, albeit some of which would likely become deposits with further exploration. In addition to the 69 identified porphyry copper deposits, 68 prospects were identified. The numbers of known deposits in the tracts ranged from none (8 tracts) to 12 (Table 1).

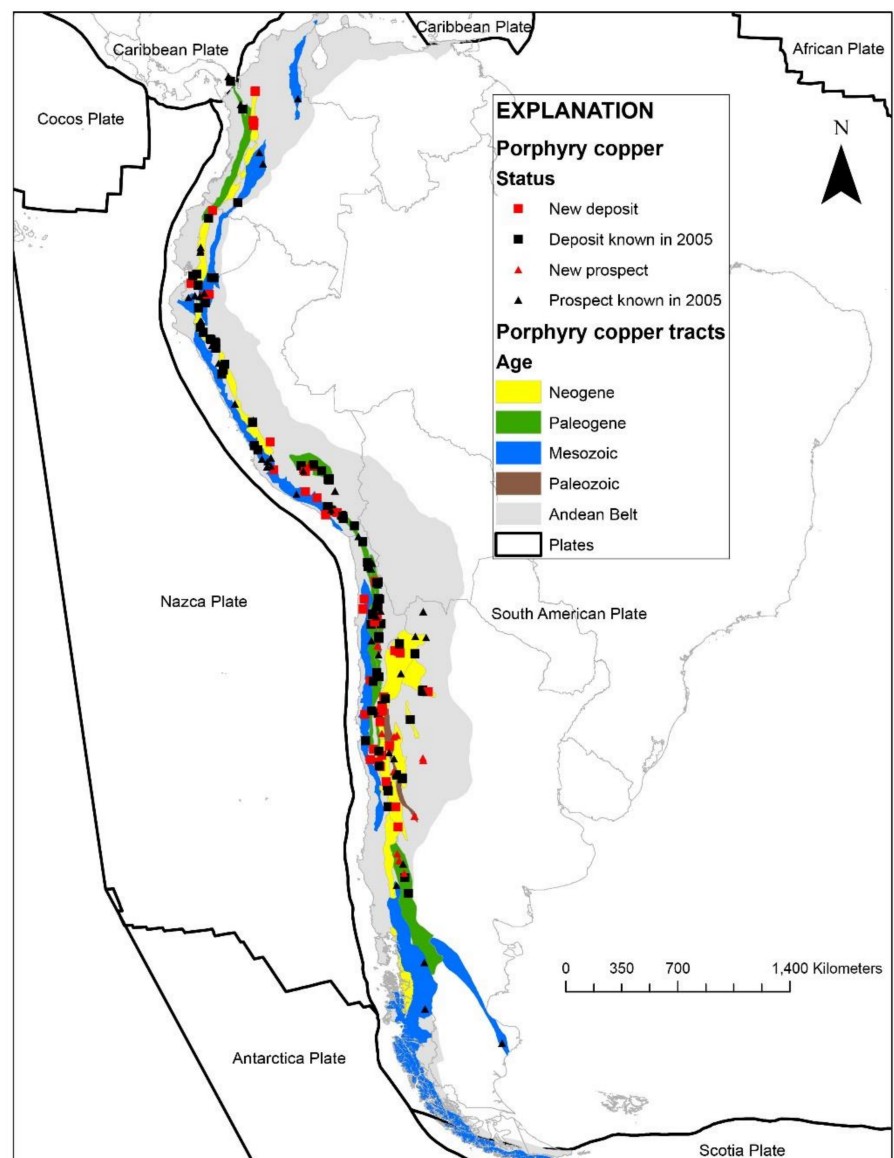

**Figure 1.** Map of South America showing the Andes region, the permissive tracts for porphyry copper deposits [5] plotted by age, locations of new deposits and prospects as of 2020, those considered in the 2005 assessment, and plate boundaries.

**Table 1.** Number of porphyry copper deposits in the Andes Mountains of South America.

| Key | Tract Number | Tract Age and Name | Number of Known Deposits in 2005 | Estimated Number of Undiscovered Deposits in 2005 | New Deposits since 2005 | Total Number of Deposits in 2020 |
|---|---|---|---|---|---|---|
| 1 | 005pCu1001 | Colombia Paleocene–Eocene Acandi | 2 | 9.6 | 1 | 3 |
| 2 | 005pCu1002 | Colombia Jurassic California | 0 | 2.9 | 3 | 3 |
| 3 | 005pCu1003 | Colombia-Ecuador-Peru Jurassic San Carlos | 5 | 12 | 1 | 6 |
| 4 | 005pCu1004 | Colombia Cretaceous Infierno Chili | 0 | 2.2 | 0 | 0 |
| 5 | 005pCu1005 | Colombia-Ecuador Miocene Chaucha | 4 | 12 | 4 | 8 |
| 6 | 005pCu1006 | Peru-Ecuador middle–late Miocene La Granja | 12 | 15 | 0 | 12 |

**Table 1.** *Cont.*

| Key | Tract Number | Tract Age and Name | Number of Known Deposits in 2005 | Estimated Number of Undiscovered Deposits in 2005 | New Deposits since 2005 | Total Number of Deposits in 2020 |
|---|---|---|---|---|---|---|
| 7 | 005pCu1007 | Peru-Ecuador Cretaceous Almacen | 2 | 3.8 | 4 | 6 |
| 8 | 005pCu1008 | Chile-Peru Paleocene–Eocene Toquepala | 12 | 12 | 2 | 14 |
| 9 | 005pCu1009 | Peru Eocene–Oligocene Antapaccay | 6 | 5.4 | 3 | 9 |
| 10 | 005pCu1010a,b | Chile Eocene–Oligocene Chuquicamata | 10 | 6.0 | 8.0 | 18 |
| 11 | 005pCu1011 | Argentina Eocene–Oligocene Taca Taca Bajo | 1 | 1.3 | 1 | 2 |
| 12 | 005pCu1012 | Chile-Argentina Eocene-Oligocene La Fortuna | 1 | 4.5 | 0 | 1 |
| 13a | 005pCu1013a | Argentina-Chile Miocene–Pliocene Cerro Casale | 1 | 11 | 7 | 8 |
| 13b | 005pCu1013b | Argentina-Chile Miocene–Pliocene Los Pelambres | 2 | 6.4 | 6 | 8 |
| 13c | 005pCu1013c | Argentina-Chile Miocene–Pliocene | 0 | 2.2 | 2 | 2 |
| 13d | 005pCu1013d | Chile-Argentina Miocene coastal | 0 | 1.3 | 1 | 1 |
| 14a | 005pCu1014a | Argentina Miocene Paramillos | 2 | 6.0 | 0 | 2 |
| 14b | 005pCu1014b | Chile Miocene–Pliocene El Teniente | 2 | 1.9 | 1 | 3 |
| 14c | 005pCu1014c | Argentina Miocene–Pliocene Bajo de la Alumbrera | 3 | 5.1 | 1 | 4 |
| 14d | 005pCu1014d | Argentina Miocene–Pliocene Nevados de Famatina | 1 | 3.5 | 0 | 1 |
| 15 | 005pCu1015 | Argentina-Chile Late Cretaceous–middle Eocene Campana Mahuida | 1 | 4.3 | 0 | 1 |
| 16 | 005pCu1016ab | Argentina Permian San Jorge | 2 | 3.5 | 0 | 2 |
| 17 | 005pCu1017 | Chile Cretaceous Antucoya | 0 | 6.7 | 6 | 6 |
| 18 | 005pCu1018 | Chile Permian El Loa | 0 | 2.2 | 1 | 1 |
| 19 | 005pCu1019 | Argentina, Late Triassic to Middle Jurassic, Bajo de la Leona | 0 | 1.6 | 0 | 0 |
| 20 | 005pCu1020 | Chile-Argentina Cretaceous Turbio | 0 | 2.3 | 0 | 0 |
| | Totals | | 69 | 145 | 51 | 120 |

Teams composed of regional experts and assessment experts considered appropriate grade and tonnage models as analogs for resource characteristics of deposits in the Andes, made probabilistic estimates of numbers of undiscovered deposits at different levels of confidence, and combined the estimates with the grade and tonnage models to simulate the in-situ metal endowment of undiscovered copper. Estimates of numbers of undiscovered deposits considered analogs with other similar areas, regional expertise, deposit density models, prospects, and potential exploration targets. Grade and tonnage characteristics for known deposits were compared with the 2005 global porphyry copper model of Singer et al. [13] to test for statistically significant differences. The model was based on grade and tonnage data for 380 deposits from around the world and included both porphyry Cu-Au and porphyry Cu-Mo subtypes. The global model was applied for most of the tracts. However, a "giant" porphyry model was developed for estimating undiscovered resources in the Eocene-Oligocene Chuquicamata and Miocene-Pliocene El Teniente tracts in Chile because the 12 known deposits in those areas had significantly higher tonnages and (or) copper grades than those in the global model at the time.

Estimates of numbers of undiscovered deposits in each permissive tract were combined with the selected grade and tonnage model using a Monte Carlo simulation program called

EMINERS, a computer program that evolved from a simulator developed by Root [12]. See [8] for details on the estimation and simulation procedure.

The 69 discovered porphyry copper deposits in the Andes as of 2005 contained 590 million metric tons (Mt) of copper. The authors estimated that about 145 porphyry copper deposits remained to be discovered with a mean total copper content of 750 Mt, as well as 20 Mt of molybdenum, 13,000 metric tons (t) of gold, and 250,000 t of silver [8].

Part of the Andes assessment was revisited in an overview of porphyry copper deposits in the Central Andes of Argentina [14]. That study identified 10 metallogenic belts for porphyry copper deposits that modified the USGS tracts and incorporated new data, updated the known resources for Argentina, and recalculated undiscovered resources in the revised metallogenic belts using the same methods. Their reassessment identified 74.21 Mt of copper in the 10 metallogenic belts as well as 1.71 Mt molybdenum, 2160.66 t gold, and 23876.71 t silver. Their estimate of mean undiscovered resources in those areas are: 239 Mt copper, 6.4 Mt molybdenum, 4240 t gold, and 77,900 t silver. Their comprehensive study demonstrated the importance of structural controls on porphyry emplacement and showed that progressive crustal thickening related to ridge subduction led to an eastward transition in magma composition in Argentina.

## 3. Porphyry Copper Discoveries in the Andes (2005–2020)

### 3.1. Data Sources

Multiple sources were used to search for new porphyry copper discoveries and updates to the 2005 porphyry copper deposit database for the Andes. Although the criteria for counting a deposit as "discovered" in the 2005 assessment called for a deposit to be thoroughly explored and not open in any direction, this requirement is rarely met because extensions of identified resources are important exploration targets especially in active areas such as the Andes. The availability of codified reporting standards such as Canadian National Instrument 43–101 (NI 43–101 reports) provides a reliable framework for reporting modern resource estimates. In many cases, changes in reported resources over time cannot always be readily explained. Cutoff grades for historical resources are not typically reported, cutoff grades change over time, properties merge, and reported resources for a particular project can vary by including, for example, supergene zones, or associated deposit types such as skarn or epithermal deposits.

A global porphyry copper database published in 2008 included new resource data for 7 deposits in the Andes that were either prospects or otherwise not included in the 2005 database [15]. Resource data for 19 of the deposits in the 2005 database were updated in 2008 and no changes were reported for the other deposits. In almost all cases, total contained copper increased between the 2005 and 2008 studies. Based on the 2008 updates, the total number of discovered porphyry copper deposits in the Andean region increased from 69 to 77 and the total contained copper increased from 590 to 800 Mt.

In addition to checking for subsequent resource updates to the 2005 and 2008 data, resource data for discoveries in the Andes through 2020 were compiled from NI 43-101 technical reports, company websites, the online Porter Geoconsultancy database and references therein [16], commercial databases, and journal articles documenting new deposits, e.g., [14,17].

Resource data are compiled as total resources (measured, indicated, and inferred) and reserves at the lowest reported cutoff grade (if cited) in publicly available sources. Past production data are included to consider the metal endowment of the deposit as a geologic entity; however, in many cases past production data are unavailable especially in deposits that have evolved from historical artisanal mining to major mining operations.

### 3.2. Analysis

The updated compilation of Andean porphyry copper deposits and prospects includes location, age, resource data, and references for 131 deposits with reported tonnage and copper grade [18]. Mining operations for some of the deposits considered in the 2005

assessment merged with other deposits. For example, the Ujina and Rosario deposits in Chile are now part of the Collahuasi deposit. Some sites that were considered as prospects in 2005, such as Los Azules in Argentina and El Hito in Ecuador now have CIM-compliant resources reported [19]. Updated grade and tonnage, locations, or new references are available for 59 of the original deposits in addition to the 51 new deposits. There are undoubtedly additional deposits and prospects in the pipeline given the exploration activity in the Andes region.

The new deposit and prospects database was converted to a shapefile and plotted in GIS (ArcGIS Desktop version 10.8.1, ESRI, Redlands, CA, USA) (to compare locations and age assignments with the 26 permissive tracts defined in the 2005 assessment. New deposits reported since 2005 occur throughout the Andes (Figure 1), including in Argentina (12), Chile (20), Colombia (6), Ecuador (4), and Peru (9). Most of the new deposits lie within or near the previously defined tracts. The Cretaceous deposits in the San Matias-Montiel area of Colombia lie outside of any defined tract. The Miocene (?) Pukaqaqa deposit in Peru lies about 20 km east of the La Granja tract.

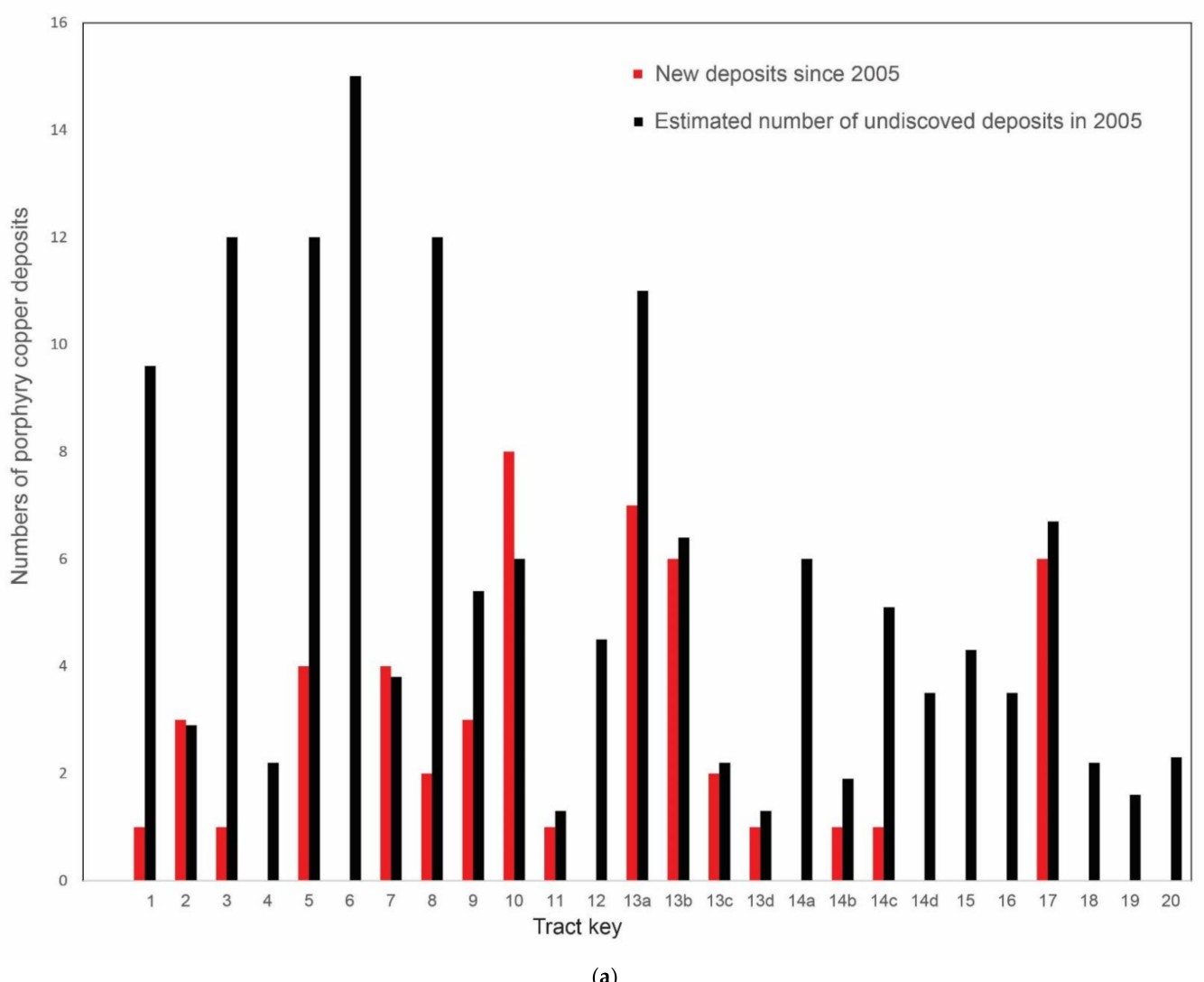

(a)

**Figure 2.** *Cont.*

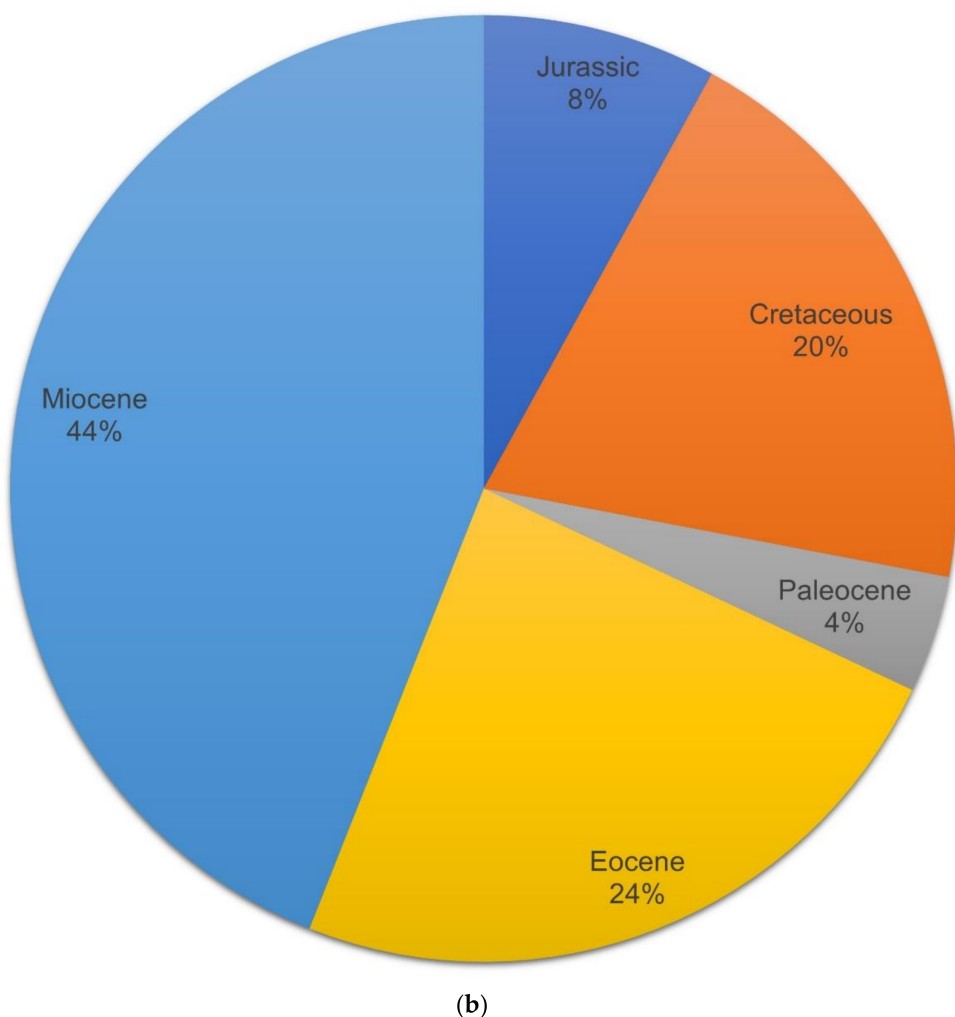

(**b**)

**Figure 2.** Porphyry copper deposits in the Andes. (**a**) Bar chart comparing estimated numbers of undiscovered deposits predicted by the 2005 assessment with numbers of deposits known in 2021 by permissive tract. See Table 1 for key to tracts. (**b**) Pie chart showing distribution of deposits known in 2021 by age.

Out of 26 tracts originally identified in the 2005 assessment, new deposits have been discovered in 4 tracts that had no known deposits in 2005 (Table 1, Figure 2a). No deposits have been discovered since the 2005 assessment in 3 tracts that had no known deposits in 2005 and no new deposits have been discovered in four tracts with previously known deposits. In some cases, the mean expected number of undiscovered deposits predicted in the 2005 assessment has been proven out. For the Andes region, the 2005 assessment predicted a total of 145 undiscovered deposits. Since that time, resource information has become available for 51 deposits, or almost a third of the predicted number of covered deposits. Many of these deposits were first discovered and explored as early as the 1970s but had not been sufficiently characterized to establish reliable resources that meet reporting standards until recently. Most of the new discoveries are in Miocene-Pliocene permissive tracts, although Eocene-Oligocene and older porphyry belts continue to host new discoveries, including the tract that hosts the world class Chuquicamata deposit (Figure 2b). The most well-endowed porphyry belts in Argentina are Miocene and the southern extension of the Miocene-Pliocene belt that hosts the giant El Teniente deposit in Chile into Argentina is a prospective underexplored region [14].

In addition to estimating numbers of undiscovered deposits, the 2005 assessment estimated amounts of copper, molybdenum, gold, and silver that could be contained in

undiscovered deposits. The results of the resource simulations are probability distributions of amounts of in situ contained metal. Selected quantiles from the assessment, along with the mean expected amount of undiscovered and the amount of mean copper that could be economic based on a simple economic filter are listed along with the amount of copper in new deposits in each tract (Table 2). For example, the Miocene Chauca tract in Colombia and Ecuador (key 5) had 4.1 million metric tons (Mt) of copper reported in 2005 with 95 percent chance of hosting 3.2 Mt or more of undiscovered copper, a 90 percent chance of 7.4 Mt, and so forth with a mean 39 Mt of which 21 Mt are likely to be economic. The economic filter data [6,7] incorporate assumptions about depth distribution (to 1 km from the surface) and infrastructure availability [10]. Comparison of the amounts of copper in new discoveries with the estimates from the assessment shows that copper has been discovered in four tracts that had no reported resources in 2005. Amounts of discovered copper in the 15 years since the assessment generally lie in the most certain quantile range (95, 90) of undiscovered copper estimated to be present. New deposits in the Eocene–Oligocene Chuquicamata tract in Chile increased the copper associated by 372.1 Mt of copper added since 2005, exceeding the mean value of 210 Mt predicted by the assessment and approaching the amount predicted at the 10 percent confidence level. Given the presence of the supergiant deposits in that tract and an exploration focus in the area, the results are not surprising. Chile leads the world in copper mine production averaging about 5.6 Mt annually in recent years with most of the production coming from the Chuquicamata-Radomiro Tomic and El Teniente mines [20]. The 490 Mt of new identified copper resources represent about 80% of the 590 Mt known in 2005. This is comparable to the 480 Mt of mean undiscovered copper that was considered to be economic. Much of the data is for inferred resources, not reserves, and resources are likely to change over time as projects move to feasibility and reserves become established.

Changes in amounts of discovered copper in porphyry deposits in the Andes on a tract basis show that copper resources in 11 of the 26 tracts have more than doubled in the 15-year period (Table 2). The assessment only dealt with copper in undiscovered deposits. An equally important consideration is resource growth over time for deposits that had identified resources reported in in 2005. Some tracts have no new discoveries but significantly increased resources from growth of deposits. For example, reported copper resources in the Eocene-Oligocene Antapaccay tract in Peru increased from 12.7 to 37.2 Mt between 2005 and 2021. As others have noted (e.g., [17]), tonnages have increased, and copper grades decreased slightly with time on a global scale (Figure 3). Deposits with grades approaching 1% Cu typically include higher grade skarn zones or enriched supergene ores.

The total copper contained in known deposits in the Andes increased from 590 Mt in 2005 to 800 Mt in 2008 to 1600 Mt as of 2021. The latest value exceeds the 10% chance of at least 1000 Mt of copper estimated for the aggregated results for South America compiled in 2015. The economic filter applied to the mean estimated in situ resources predicted that about 60 percent of those resources would be economic given the assumption applied. Resource growth in deposits that were known in 2005 (1100 Mt copper) exceeds copper resources in new discoveries (490 Mt copper) by a factor of 2 (Table 2).

**Table 2.** Copper resources in million metric tons.

| Key | 2005 Dis-covered Cu | 2020 Dis-covered Cu | Resource Growth 2005 to 2020 | Cu in New Deposits 2005 to 2020 | 2005 Probabilistic Estimates of Undiscovered Cu | | | | | | Value Class |
|-----|-----|-----|-----|-----|-----|-----|-----|-----|-----|-----|-----|
| | | | | | 95 | 90 | 50 | 10 | Mean | Economic | |
| 1 | 10.0 | 21.1 | 10.0 | 11.2 | 0.8 | 3.1 | 23.0 | 76.0 | 33.0 | 18.0 | high |
| 2 | 0 | 0 | 0 | 0 | 0 | 0.3 | 4.7 | 23.0 | 9.7 | 5.2 | medium |
| 3 | 9.0 | 15.4 | 14.4 | 1.0 | 3.1 | 7.1 | 31.0 | 85.0 | 40.0 | 20.0 | high |
| 4 | 0 | 0 | 0 | 0 | 0 | 0.2 | 3.3 | 19.0 | 7.7 | 4.4 | high |
| 5 | 4.1 | 16.2 | 10.1 | 6.1 | 3.2 | 7.4 | 30.0 | 81.0 | 39.0 | 21.0 | high |
| 6 | 46.7 | 59.1 | 59.1 | 0 | 5.8 | 11.0 | 39.0 | 100.0 | 49.0 | 24.0 | high |

**Table 2.** *Cont.*

| Key | 2005 Discovered Cu | 2020 Discovered Cu | Resource Growth 2005 to 2020 | Cu in New Deposits 2005 to 2020 | 2005 Probabilistic Estimates of Undiscovered Cu | | | | | | Value Class |
|---|---|---|---|---|---|---|---|---|---|---|---|
| | | | | | 95 | 90 | 50 | 10 | Mean | Economic | |
| 7 | 0.6 | 5.2 | 0.7 | 4.6 | 0 | 0.3 | 6.8 | 33.0 | 14.0 | 7.7 | low |
| 8 | 55.4 | 117.5 | 114.7 | 2.8 | 4.0 | 7.9 | 33.0 | 92.0 | 43.0 | 27.0 | high |
| 9 | 12.7 | 37.2 | 24.8 | 12.4 | 0.8 | 2.1 | 11.0 | 44.0 | 19.0 | 9.9 | high |
| 10 | 252.0 | 855.5 | 483.4 | 372.1 | 22.0 | 46.0 | 190.0 | 400.0 | 210.0 | 170.0 | high |
| 11 | 3.0 | 11.8 | 2.9 | 0.2 | 0 | 0 | 1.0 | 10.0 | 4.2 | 2.6 | high |
| 12 | 3.0 | 11.5 | 11.5 | 0 | 0.2 | 1.1 | 8.5 | 34.0 | 15.0 | 10.0 | high |
| 13a | 2.9 | 22.5 | 4.2 | 18.3 | 1.8 | 4.2 | 25.0 | 89.0 | 38.0 | 20.0 | high |
| 13b | 25.6 | 89.7 | 54.0 | 35.7 | 0.8 | 2.6 | 14.0 | 50.0 | 22.0 | 12.0 | high |
| 13c | 0 | 3.6 | 0 | 3.6 | 0 | 0.2 | 3.3 | 18.0 | 7.7 | 4.0 | high |
| 13d | 0 | 1.0 | 0 | 1.0 | 0 | 0 | 1.0 | 11.0 | 4.5 | 2.5 | low |
| 14a | 2.1 | 2.1 | 2.1 | 0.0 | 0.3 | 1.1 | 11.0 | 50.0 | 21.0 | 11.0 | high |
| 14b | 148.6 | 277.2 | 267.5 | 9.7 | 0 | 7.6 | 49.0 | 150.0 | 69.0 | 56.0 | high |
| 14c | 12.3 | 16.5 | 15.9 | 0.6 | 0.8 | 2.2 | 11.0 | 38.0 | 17.0 | 8.9 | high |
| 14d | 1.1 | 1.1 | 1.1 | 0 | 0 | 0.3 | 6.1 | 29.0 | 12.0 | 6.3 | high |
| 15 | 1.0 | 1.0 | 1.0 | 0 | 0 | 0.2 | 5.9 | 40.0 | 15.0 | 8.6 | high |
| 16 | 1.9 | 1.3 | 1.3 | 0 | 0.2 | 0.8 | 6.2 | 27.0 | 12.0 | 6.6 | medium |
| 17 | 0 | 9.3 | 9.3 | 9.3 | 1.0 | 2.6 | 15.0 | 52.0 | 23.0 | 15.0 | high |
| 18 | 0 | 3.1 | 3.1 | 3.1 | 0 | 0.2 | 3.2 | 18.0 | 7.5 | 4.8 | high |
| 19 | 0 | 0 | 0 | 0 | 0 | 0 | 1.4 | 15.0 | 5.9 | 2.8 | very low |
| 20 | 0 | 0 | 0 | 0 | 0 | 0 | 3.0 | 20.0 | 7.8 | 3.9 | very low |
| Total | 590 | 1600 | 1100 | 490 | - | - | - | - | 750 | 480 | - |

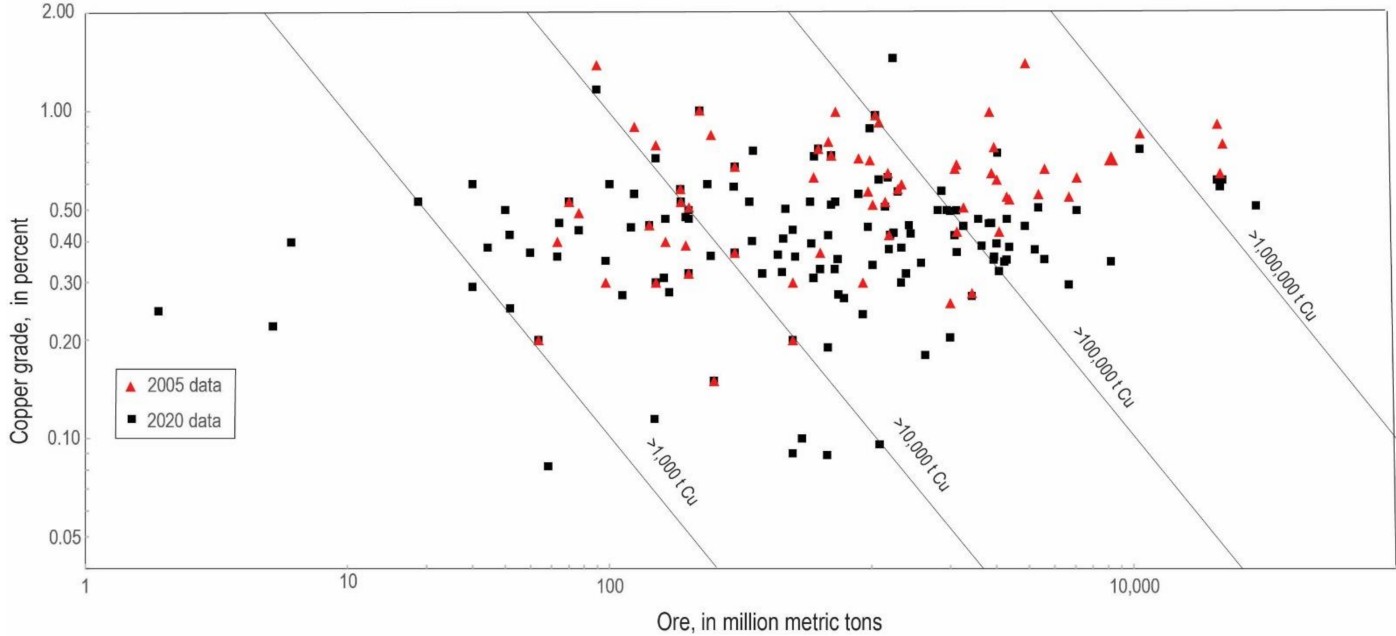

**Figure 3.** Scatter plot of average copper grade versus ore tonnage for porphyry copper deposits known in the Andes in 2005 and 2020.

## 4. Discussion

Tracts that outline Mesozoic age rocks in southern Chile and Argentina and lack known porphyry copper deposits represent areas where the relative amount of exposed intrusive rocks greatly exceeds the amount of volcanic rocks suggesting that levels of exposure may be too deep to preserve porphyry copper deposits (Figure 4a). Alternatively,

these areas may simply be underexplored. The 2005 assessment excluded Cretaceous large batholiths in central Colombia from the Infierno Chili tract as too deeply eroded. Infierno Chili is a porphyry copper prospect hosted in the 131 Ma Ibagué batholith. The Cretaceous porphyry copper deposits recently discovered in the San Matias-Montiel area (Figure 4b) including Alacran, Montiel East, Montiel West, and Costa Azul were not included in the 2005 assessment. These may represent a more prospective area than previously thought. These younger (73 to 77 Ma) Au-rich Cretaceous porphyry systems [21,22] in the Western Cordillera of Colombia represent island arc-type deposits that occur in accreted oceanic terrane rather than the continental arc geologic settings that characterize most of the Andes [23]. Most of the porphyry copper deposits in the Andes contain between 1 and 100 Mt of copper, although the giant deposits (>100 Mt copper) such as El Teniente and Chuquicamata (Figure 4b) continue to expand even as production proceeds (Figure 4b). Molybdenum grades and gold grades are reported for 60 and 49 deposits, respectively (Figure 4c,d). Molybdenum-rich deposits are most prevalent in the region of the supergiant deposits in the central Andes whereas gold-rich deposits occur in both the central and northern Andes (Figure 4d). Silver is reported for 49 deposits but not always along with gold. A few deposits report the presence of the critical minerals palladium (~35 ppb) and platinum (8 ppb) in concentrates.

Comparison of new discoveries with the Andes assessment show that the 2005 predictions on a tract-by-tract basis were neither wildly speculative nor conservative but generally captured the most likely (50% to 95% chance) outlook for undiscovered resources in the area. The assessment failed to anticipate significant new discoveries in the Chuquicamata (tract 10) and Los Pelambres (tract 13a) tracts where the amount of additional copper resources exceeds the predicted mean expected values for the tracts. Assessment results provide a tool for anticipating future sources of mineral resources and cannot be considered "right" or "wrong". Only complete exploration in three dimensions could establish the full endowment of a region. As new geologic and exploration data become available, assessments can be revised and updated. New digital geologic maps of the region [24,25], application of prospectivity mapping techniques [26], and new tools for quantitative mineral resource assessment [27,28] and economic analysis [29] are now available for exploration and assessment. Many companies release geophysical surveys and other data on websites and in technical reports that were not available in 2005. The assessment only considered deposits in the upper 1 km of the Earth's surface; mining to deeper depths is becoming increasingly technologically feasible and a number of deposits are moving from open pit to underground mining to extend mine life such as the ongoing development at Chuquicamata which is expected to extend mine life by 40 years [20].

The Eocene–Oligocene Chuquicamata tract in Chile has the largest recent contained copper resource (855 Mt) and the largest amount of copper added since 2005 (372 Mt) yet the tract area represents only 2% of the entire area assessed (1.2 million km$^2$). There is no relationship between the amount of copper and tract size. The tracts that were ranked as having high expected value based on the economic filter [7] are tracts that have had new discoveries and increases in identified resources (Figure 5). The tracts ranked as having medium economic potential had no new discoveries and two tracts ranked as having low potential had new discoveries. Analysis of permissive tracts that have been assessed using the three-part form of quantitative assessment showed discovery order is not a reliable predictor of deposit size unlike the oil industry where larger oil pools tend to be discovered earlier in exploration [30]. The discovery order study analyzed discovery date and deposit size for the 26 permissive tracts defined for the 2005 Andes assessment [8]. The twelve deposits known at that time in Paleocene-Eocene Toquepala tract SA08 (Table 1, key 8) were partly exposed; most of the predicted deposits were expected to be under post-mineral cover that comprised 70% of the tract area. The two new deposits that have reported resources since that time (Los Calatos and Mollucas) do in fact have extensive cover, oxide, and secondary sulfide zones overlying deep (500 m) primary sulfide zones identified by drilling, and surface exposure mainly expressed as alteration.

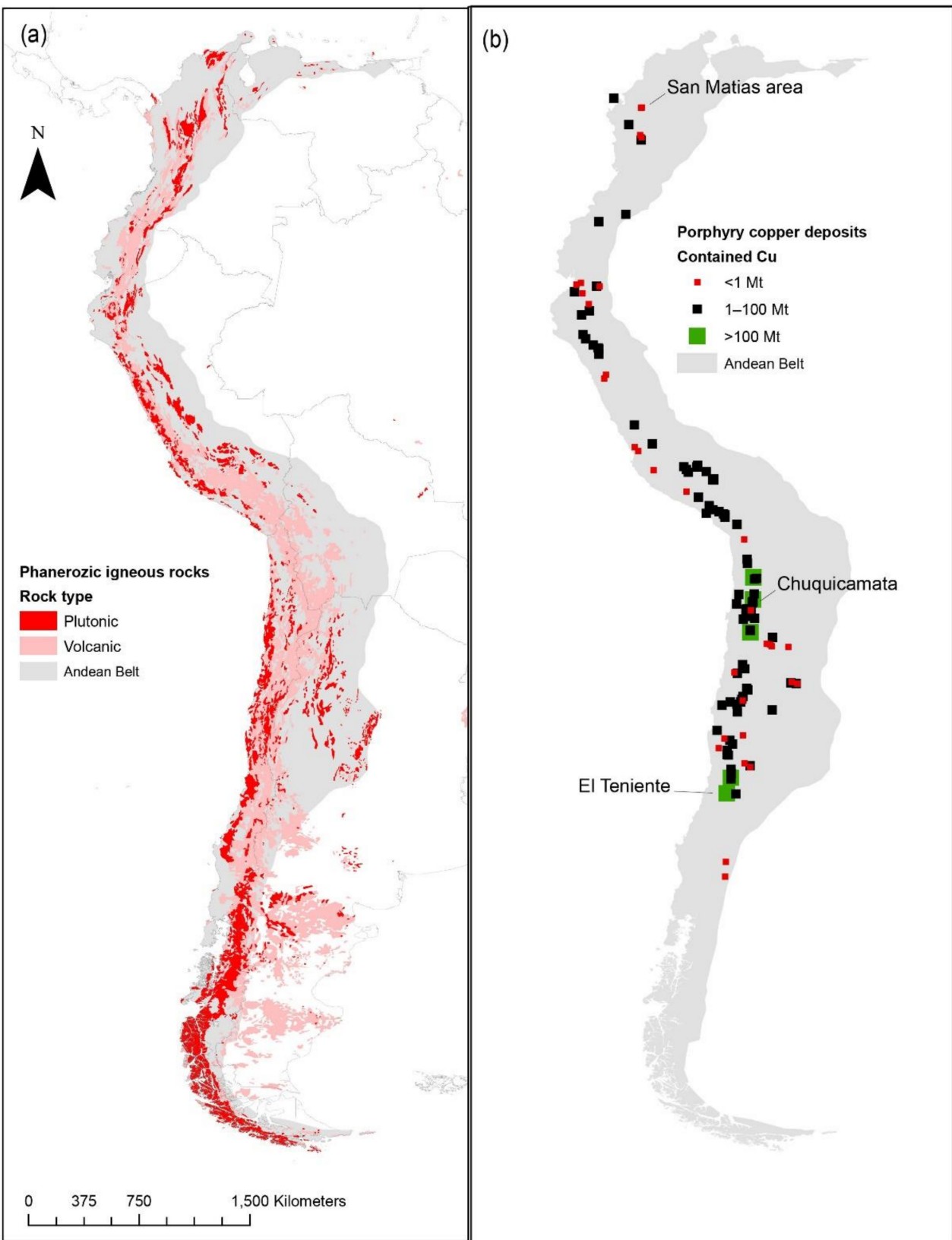

**Figure 4.** *Cont.*

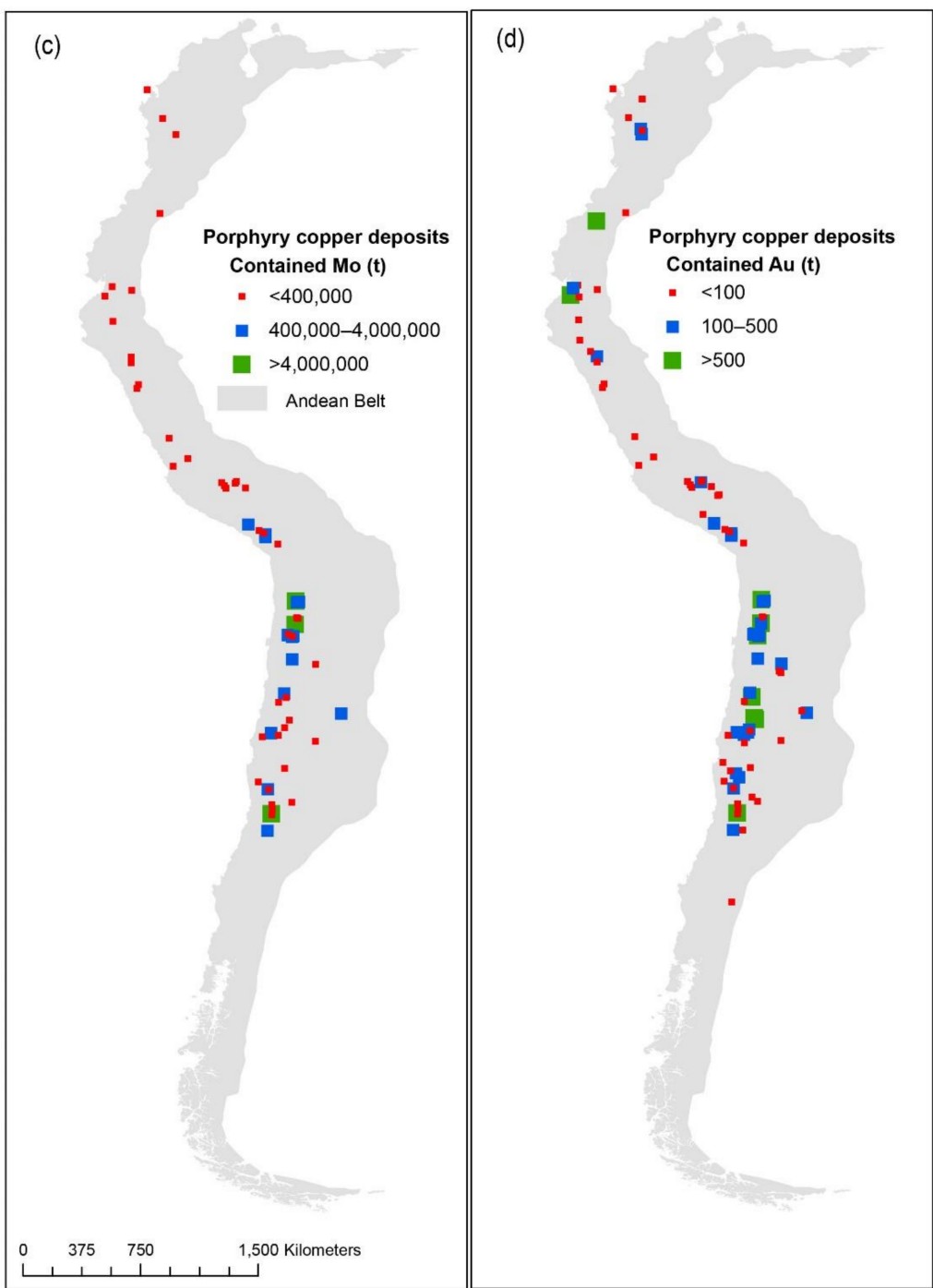

**Figure 4.** Maps showing the distribution of igneous rocks and contained metals in porphyry copper deposits. (**a**) Phanerozoic igneous rocks, (**b**) contained copper, (**c**) contained molybdenum, and (**d**) contained gold. Igneous rocks are from Tapias et al. [25].

Resource growth in known deposits typically is not considered in mineral resource assessment of undiscovered deposits. However, in areas of active mining such as the Andes, where continued exploration expands mine life and technology allows conversion from open pit to underground mining to tap deep resources, resource growth may be as, or more significant, than greenfields additions to global copper resources. Many of the recently discovered deposits with <1 million metric tons of copper are open in one or more directions, are not fully delineated in terms of resources, and are likely to expand as development proceeds.

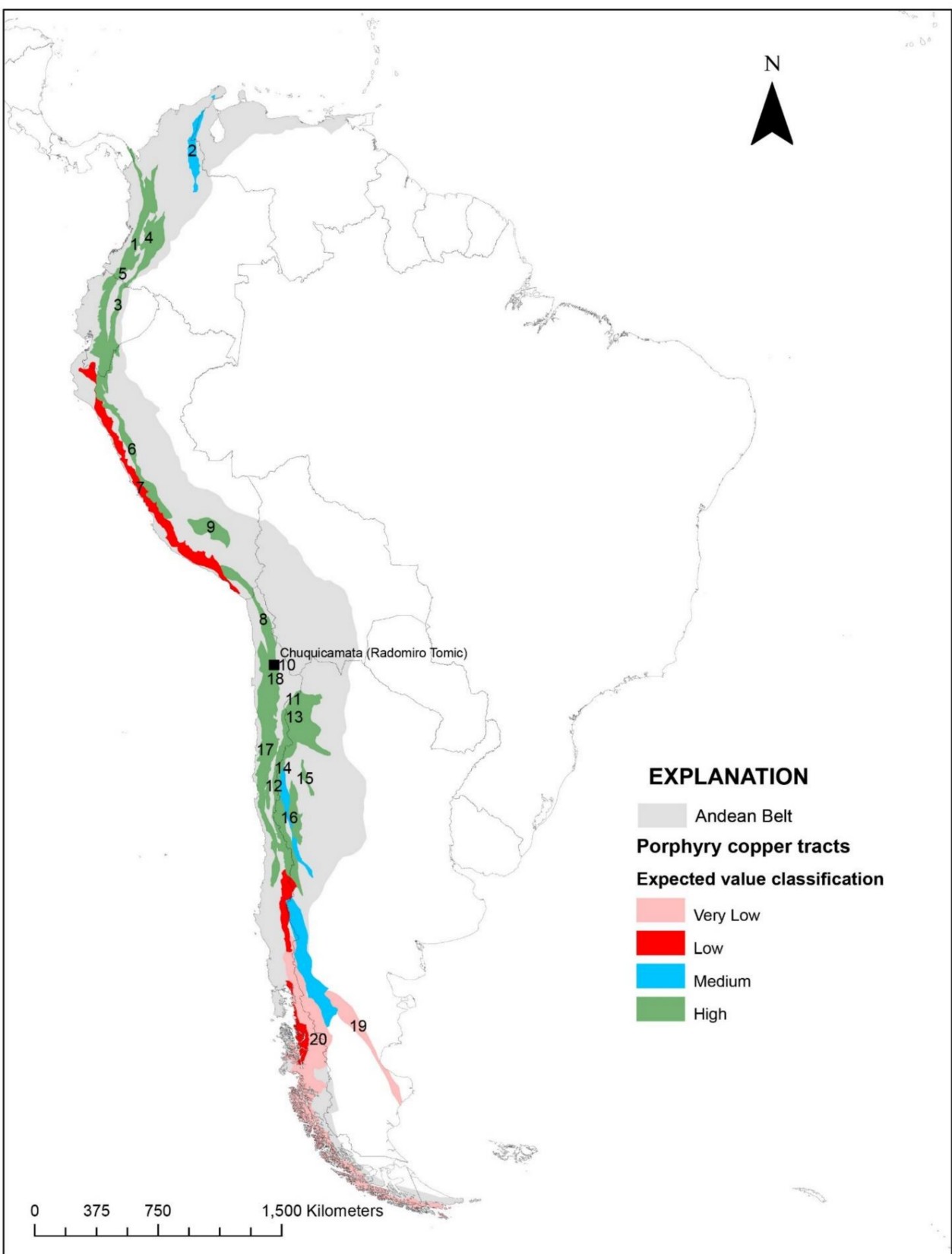

**Figure 5.** Map showing the distribution of permissive tracts for porphyry copper deposits by expected value classification. Tracts are from Cunningham et al. [8]; expected values are from Hammarstrom et al. [4]. See Table 1 for key to tract numbers.

Exploration is episodic and subject to many political, societal, and economic constraints. Much of the recent exploration, especially in the northern Andes, is focused on gold in epithermal systems, some of which contain low-grade copper and probably overlie deeper porphyry systems that remain to be drilled out. However, many areas remain incompletely explored and even well-explored areas continue to host new discoveries. In addition to ore, the vast amounts of mine tailings in the Andes could contain additional unrecovered resources including critical minerals such as palladium, platinum, and rhenium that could be recovered as part of reclamation efforts if technically and economically viable to do so. Companies rarely report the complete chemistry of ores and concentrates. Ongoing efforts to provide complete geochemical characterization of ore samples and tailings from different deposit types, including porphyry copper deposits, will help close that data gap and could lead to more efficient and widespread recovery of critical minerals [31–34].

**Funding:** This research was funded by the Mineral Resources Program of the U.S. Geological Survey.

**Supplementary Materials:** The following are available online at https://www.mdpi.com/article/10.3390/min12070856/s1, Table S1: Porphyry copper deposits in the Andes—Analysis of resource data including age, location, comments, and references.

**Data Availability Statement:** The data presented in this study and included in the Supplementary Table S1 are openly available [18].

**Acknowledgments:** Floyd Gray, USGS, provided helpful comments in a review of a draft of this manuscript. E.G. Boyce, USGS, facilitated data compilation by providing a collection of NI43-101 reports for South American deposits. Three anonymous reviewers provided useful comments for improving the manuscript.

**Conflicts of Interest:** The author declares no conflict of interest.

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
