# Peer review of "Porphyry Copper: Revisiting Mineral Resource Assessment Predictions for the Andes"

_minerals, doi:10.3390/min12070856_

Round 1

Reviewer 1 Report

The submitted manuscript is a well-written contribution of the porphyry copper system in South America. The Author gave a logic summary as well as review of the economical impact of the largest copper deposit from all around the world. However, there are some major as well as minor suggestions, which must be considered prior to the publication of the manuscript. I reccommend the paper for the publication with a minor revision.

Major Suggestions

Introduction

Define the porphyry copper systems of the Earth in a separated section, as follows: age and geographical distribution, importance as a natural resource, associated minerals (e.g., Mo), economic impact, etc.

Give examples other large porphyry copper systems including Malanjkhand, India (Sikka and Nehru, 1997), Tongkuangyu, China (Weixing and Dazhong, 1987), Boddington, Australia (Roth et al., 1991), Haib, Namibia (Minnitt, 1986) and Troilus, Quebec (Fraser, 1993). 

Literature Suggestion: 

Sikka, D.G. and Nehru, C.E., 1997, Review of Precambrian porphyry Cu±Mo±Au deposits with special reference to Malanjkhand porphyry copper deposit, Madhya Pradesh, India: Journal of the Geological Society of India, v. 49, p. 239-288.

Weixing, Hu and Dazhong, Sun., 1987, Mineralization and evolution of the Early Proterozoic copper deposits in the Zhongtiao Mountains: Acta Geologica Sinica, v. 61, p. 152-165.

Roth, E., Groves, D.E., Anderson, G., Daley, L. and Staley, R., 1991, Primary mineralization at the Boddington mine, Western Australia: An Archean porphyry Cu-Au-Mo deposit; in Ladeira, E.A. , ed., Brazil Gold 91, The Economic Geology, Geochemistry and Genesis of Gold Deposits: A.A. Balkema, Rotterdam, p. 481-488.

Minnitt, R.C.A., 1986, Porphyry copper mineralization, Haib River, Southwest Africa, Namibia; in Anhaeusser, C.R., and Maske, S., eds., Mineral Deposits of Southern Africa, v. II: The Geological Society of South Africa, p. 1567-1585. 

Fraser, R.J., 1993, The Lac Troilus gold-copper deposit, northwestern Quebec: a possible Archean porphyry system: Economic Geology, v. 88, p. 1685-1699.

Minor Suggestions:

1. Introduction:

Page 1, Line 7: delete the space between "The" and  "study"

Page 2, Line 48 "....economically recoverable [7,4]."-numbering: "[4,7]"  

2. Background:

Page 3, Line 92:  "...Singer and others (2005)..."It must be in left and right square brackets, as follows. Singer et al. [10].  

Page 4, Line 111: Use "Table 2" instead of "table 2"

Page 6, Line 159: delete space between "part of" and "the Collahuasi"

Discussion

Page 11, Line 275: "earth's" Capital "E"

Author Response

Introduction: I added a few sentences on porphyry copper deposits and 3 new references, as suggested by the reviewers.   I do not think a longer discussion of porphyry copper deposits is warranted in this paper because  the topic of the paper is mineral resource assessment. 

I corrected all of the other  items noted by the reviewer.

Reviewer 2 Report

The paper is an interesting review study of a mineral resource assessment of copper deposits in South America. The paper provides interesting information and is conceptually well managed. I have no major comments on the content.

There are only minor points that need to be corrected: 

1) Fig. 2 - the second category of the legend does not have a complete number/year (should be 2021)

2) Figure 4.c has an incorrectly defined interval in the legend (4,000,000-4,000,000)

Translated with www.DeepL.com/Translator (free version)

Author Response

Thank you for noting the issues with figures 2 and 4. Figure 2 was cut off on the right side. The wording  is correct - the chart compares numbers of deposits discovered since 2005 with numbers of undiscovered deposits that were predicted in the 2005 assessment. 

Reviewer 3 Report

The author has done an excellent job on this ms, and it should be published.

Author Response

Thank you. No revisions requested by this reviewer.